# Virus-Induced Plant Volatiles Promote Virus Acquisition and Transmission by Insect Vectors

**DOI:** 10.3390/ijms24021777

**Published:** 2023-01-16

**Authors:** Xuefei Chang, Yating Guo, Yijia Ren, Yifan Li, Fang Wang, Gongyin Ye, Zhaozhi Lu

**Affiliations:** 1Shandong Engineering Research Center for Environment-Friendly Agricultural Pest Management, College of Plant Health and Medicine, Qingdao Agricultural University, Qingdao 266109, China; 2State Key Laboratory of Rice Biology & Ministry of Agriculture and Rural Affairs Key Laboratory of Molecular Biology of Crop Diseases and Insects, Institute of Insect Sciences, Zhejiang University, Hangzhou 310058, China

**Keywords:** *Nephotettix virescens*, rice dwarf virus, plant volatiles, olfactory behavior, virus dispersal

## Abstract

Rice dwarf virus (RDV) is transmitted by insect vectors *Nephotettix virescens* and *Nephotettix cincticeps* (Hemiptera: Cicadellidae) that threatens rice yield and results in substantial economic losses. RDV induces two volatiles ((E)-β-caryophyllene (EBC) and 2-heptanol) to emit from RDV-infected rice plants. However, the effects of the two volatiles on the olfactory behavior of both non-viruliferous and viruliferous *N. virescens* are unknown, and whether the two volatiles could facilitate the spread and dispersal of RDV remains elusive. Combining the methods of insect behavior, chemical ecology, and molecular biology, we found that EBC and 2-heptanol influenced the olfactory behavior of non-viruliferous and viruliferous *N. virescens*, independently. EBC attracted non-viruliferous *N. virescens* towards RDV-infected rice plants, promoting virus acquisition by non-viruliferous vectors. The effect was confirmed by using *oscas*1 mutant rice plants (repressed EBC synthesis), but EBC had no effects on viruliferous *N. virescens*. 2-heptanol did not attract or repel non-viruliferous *N. virescens*. However, spraying experiments showed that 2-heptanol repelled viruliferous *N. virescens* to prefer RDV-free rice plants, which would be conducive to the transmission of the virus. These novel results reveal that rice plant volatiles modify the behavior of *N. virescens* vectors to promote RDV acquisition and transmission. They will provide new insights into virus–vector–plant interactions, and promote the development of new prevention and control strategies for disease management.

## 1. Introduction

Plant viruses are most often transmitted by insect vectors including thrips, whiteflies, planthoppers, leafhoppers, and aphids [1], and they seriously threaten crop yield and lead to major economic losses [2]. Insect vectors, plant viruses, and host plants form complex multilevel interactions that affect virus dispersal [3]. For persistent plant viruses transmitted by insect vectors, studies have found that non-viruliferous insect vectors initially preferred virus-infected plants, but once the virus was acquired by the vectors, they showed a predilection for virus-free plants, and this phenomenon may accelerate the outbreak of virus [4]. Changes in host preference by insect vectors between virus-free and virus-infected plants after virus acquisition were first reported in the wheat–*Rhopalosiphum padi*–barley yellow dwarf virus (BYDV) pathosystem [5] and subsequently in several other pathosystems [6,7,8,9]. These studies indicated that changes in the host preference of insect vectors after virus acquisition might be associated with plant volatiles induced by virus infection [4,10].

Host plants release several different kinds of volatile organic compounds (VOCs) during infection by plant viruses including terpenes, sesquiterpenes, green leaf volatiles (GLVs), fatty acid derivatives, aromatics, and nitrogen-containing compounds, as well as the volatile plant hormones, methyl salicylate, and methyl jasmonate [11]. VOCs induced by plant viruses have different roles in the olfactory behavior of insect vectors such as attractions and repellents, or they have no effects [11]. For instance, VOCs such as GLVs, sesquiterpenes, and terpenoids induced by cucumber mosaic virus (CMV), BYDV, and potato leaf roll virus (PLRV) attracted non-viruliferous insect vectors to feed virus-infected plants over virus-free plants [6,12,13]. *Myzus persicae* and *Aphis glycines* preferred CMV-infected pepper plants to CMV-free pepper plants, due to the CMV-infected pepper plants emitting more ethylene than CMV-free pepper plants [14].

Previous studies have also found that insect vectors performed better on virus-infected host plants, and this was the result of lower volatile emissions from the virus-infected plants [15,16,17,18,19]. For example, CMV and tomato yellow leaf curl China virus (TYLCCNV) inhibit the emission of terpenes that are repellent to insect vectors from virus-infected plants, which is conducive to the feeding behavior of the vector insects [18,20]. However, in the past several years, most studies paid more attention to the behavior responses of insect vectors to mixed VOCs from infected plants, artificial blends that mimic the natural blends, or individual compounds prominent in those blends [4,11]; the effects of individual VOCs on the olfactory behavior of both non-viruliferous and viruliferous insect vectors were not largely studied.

Rice dwarf virus (RDV), the causal agent of rice dwarf disease, belongs to the genus *Phytoreovirus*, family *Reoviridae* [21]. *Nephotettix virescens* (Hemiptera: Cicadellidae) is one of the main insect vectors other than *Nephotettix cincticeps*, which could transmit RDV to rice plants in a persistent-propagative manner [21,22]. RDV infection could inhibit rice growth and cause severe symptoms such as white chlorotic spots on rice leaves, shorter and fewer roots, and increased tiller number, consequently decreasing grain yield [23,24]. Once widespread infection occurs, it is hard to control the disease [25]. In our previous study, we showed that (E)-β-caryophyllene (EBC) and 2-heptanol, induced by RDV infection, independently influenced the olfactory behavior of non-viruliferous and viruliferous *N. cincticeps* [26]. However, the effects of the two VOCs on the olfactory behavior of the important virus vector *N. virescens* are largely unexplored, and whether the two individual VOCs can promote RDV acquisition and transmission remain elusive. In the present study, we hypothesized that EBC and 2-heptanol mediate the feeding and plant odor preference of the insect vector *N. virescens*, and consequently facilitate RDV acquisition and transmission by *N. virescens*. Herein, a series of complementation assays were designed to determine our hypothesis. This study extends our knowledge of virus–insect vector–host plant interactions and may provide an important step in developing strategies based on the production of rice plants with altered VOC profiles.

## 2. Results

### 2.1. Rice Volatiles Influence the Selection Preference of Non-Viruliferous and Viruliferous N. Virescens

The feeding preferences of *N. virescens* between WT (RDV-free) and WT-RDV (RDV-infected) rice plants were recorded for 72 h. For non-viruliferous insects, no clear feeding preference was found between the two different rice plants during the first 8 h post-inoculation (PI) (other than 4 h) (2 h, P = 0.902; 4 h, P = 0.046; 8 h, P = 0.064). From 24 h PI, the non-viruliferous insects clearly preferred WT-RDV rice plants compared to WT rice plants (24 h, P = 0.007; 48 h, P = 0.005; 72 h, P = 0.008) (Figure 1A). Similarly, during the first 4 h PI, viruliferous insects showed no significant preferences between the two different rice plants (2 h, P = 0.521; 4 h, P = 0.582); however, from 8 h PI, viruliferous insects preferred to feed on WT rice plants to WT-RDV rice plants (8 h, P = 0.021; 24 h, P = 0.041; 48 h, P = 0.024; 72 h, P = 0.006) (Figure 1B).

To verify the involvement of rice volatiles mediating the olfactory behavior of *N. virescens*, WT (RDV-free) and WT-RDV (RDV-infected) rice plant odors were applied via two opposite arms of a four-chamber olfactometer. Results showed that the time non-viruliferous *N. virescens* spent in the four arenas was significantly different (Figure 2A, χ^2^ = 60.816, *p* < 0.001, *n =* 41). The time non-viruliferous *N. virescens* invested in the arena containing WT-RDV plant odors was almost 569.3 s, whilst in the arena containing WT plant odors it was almost 239.7 s. The tracks of non-viruliferous *N. virescens* in the four arenas confirmed this discovery (Appendix A). These findings suggested that non-viruliferous *N. virescens* preferred WT-RDV plant odors over WT plant odors. The residence time of viruliferous *N. virescens* in the four arenas also showed significant differences (Figure 2B, χ^2^ = 67.936, *p* < 0.001, *n =* 44). Viruliferous *N. virescens* remained for almost 484.4 s in the WT plant odor area, and almost 270.8 s in the WT-RDV plant odor area. The walking path of viruliferous *N. virescens* in the four arenas is shown in Appendix A. These results demonstrated that viruliferous *N. virescens* showed a significant preference for WT plant odors over WT-RDV plant odors. All these findings confirmed that rice volatiles influence the selection preferences between WT and WT-RDV rice plants of non-viruliferous and viruliferous *N. virescens*.

### 2.2. EBC Was Attractive to Non-Viruliferous N. Virescens, while 2-Heptanol Was Repellent to Viruliferous N. Virescens

To illustrate the two rice VOCs induced by RDV infection could affect the feeding and plant odor preferences of non-viruliferous and viruliferous *N. virescens*, for the first, a four-quadrant olfactometer should first be used to determine the effects of standard VOCs on the olfactory behavior of *N. virescens* (as described above). The time non-viruliferous *N. virescens* spent in the arena permeated with EBC was significantly longer than in the other three arenas at any tested concentration (Figure 3A, 0.01 μg μL^−1^, χ^2^ = 54.229, *p* < 0.001, *n =* 42; 0.1 μg μL^−1^, χ^2^ = 68.800, *p* < 0.001, *n =* 42; 1 μg μL^−1^, χ^2^ = 54.086, *p* < 0.001, *n =* 42). The time viruliferous *N. virescens* spent in the four arenas was not significantly different at any of the concentrations tested (Figure 3B, 0.01 μg μL^−1^, χ^2^ = 0.930, P = 0.818, *n =* 40; 0.1 μg μL^−1^, χ^2^ = 1.350, P = 0.717, *n =* 40; 1 μg μL^−1^, χ^2^ = 3.031, P = 0.387, *n =* 39). The walking tracks of *N. virescens* exposed to EBC at 0.1 μg μL^−1^ are shown in Appendix A. These results suggest that EBC was attractive to non-viruliferous *N. virescens*, but had no effect on viruliferous *N. virescens*.

Behavioral assays revealed that 2-heptanol was not attractive or repellent to non-viruliferous *N. virescens* at any of the three dosages tested (Figure 3C, 0.01 μg μL^−1^, χ^2^ = 4.832, P = 0.185, *n =* 38; 0.1 μg μL^−1^, χ^2^ = 0.270, P = 0.966, *n =* 40; 1 μg μL^−1^, χ^2^ = 1.215, P = 0.749, *n =* 41). However, 2-heptanol was significantly repellent to viruliferous *N. virescens* at the three dosages tested (Figure 3D, 0.01 μg μL^−1^, χ^2^ = 58.304, *p* < 0.001, *n =* 36; 0.1 μg μL^−1^, χ^2^ = 44.367, *p* < 0.001, *n =* 36; 1 μg μL^−1^, χ^2^ = 82.395, *p* < 0.001, *n =* 37). The walking paths of *N. virescens* are shown in Appendix A (2-heptanol, 0.1 μg μL^−1^).

### 2.3. Confirming the Attraction of EBC to Non-Viruliferous N. Virescens

Previous research showed that oscas1 rice plants infected with RDV could not emit more EBC [26]. Therefore, we used oscas1 mutant rice plants to verify the attraction of EBC to non-viruliferous *N. virescens* with feeding and plant odor preference studies. There was no significant preference for feeding selection by non-viruliferous *N. virescens* between WT (RDV-free) and oscas1 (RDV-free) rice plants, or between oscas1 (RDV-free) and oscas1-RDV (RDV-infected) rice plants during the entire investigation (Figure 4A,B, *p* > 0.05). In contrast, the number of non-viruliferous vectors was more abundant on WT-RDV (RDV-infected) rice plants than that on oscas1-RDV (RDV-infected) rice plants (except for 2 h PI, P = 0.475) (Figure 4C).

To further verify the attraction of EBC to non-viruliferous *N. virescens*, three plant odor preference assays were conducted. When rice volatiles from WT (RDV-free) and oscas1 (RDV-free) were added via two opposite arenas of the olfactometer, the time non-viruliferous vectors spent in the olfactometer was significantly different (Figure 5A, χ^2^ = 27.000, *p* < 0.001, *n =* 38), but no significant difference between the two odors was found (*p* > 0.05). However, there was no significant preference by non-viruliferous vectors between oscas1 (RDV-free) and oscas1-RDV (RDV-infected) rice plant odors (Figure 5B, χ^2^ = 6.692, P = 0.082, *n =* 39). Non-viruliferous vectors walked more in the WT-RDV (RDV-infected) plants odor arena compared to the other three odor arenas (oscas1-RDV (RDV-infected) and two clean air odor fields) (Figure 5C, χ^2^ = 22.573, *p* < 0.001, *n =* 40). The walking tracks of non-viruliferous vectors in a four-field olfactometer with the three different assays can be found in Appendix A–C.

### 2.4. Verifying the Repellency of 2-Heptanol to Viruliferous N. Virescens

Individual VOC assays showed that 2-heptanol was repellent to viruliferous *N. virescens*. To further confirm the role of 2-heptanol in the behavior of viruliferous vectors, spraying experiments were conducted. There was no significant preference by viruliferous *N. virescens* for feeding on WT rice plants treated with pure lanolin paste (control plants) or WT rice plants treated with synthetic 2-heptanol in lanolin paste (treated plants) during the first 4 h PI (Figure 6A, 2 h, P = 0.858; 4 h, P = 0.064). However, viruliferous *N. virescens* preferred feeding on the control plants compared to treated plants at 8 h PI (Figure 6A, P = 0.007), and this performance was throughout the rest duration (24 h, P = 0.005; 48 h, P = 0.007; 72 h, P = 0.005).

The time viruliferous vectors spent in the control plant odor arena was significantly longer than that in the treated plant odor arena (Figure 6B, χ^2^ = 40.129, *p* < 0.001, *n =* 34), and the walking tracks of viruliferous vectors can be seen in Appendix A.

### 2.5. EBC and 2-Heptanol Promote the RDV Acquisition and Transmission

Two combinations of rice plants (WT-RDV vs. WT; oscas1 vs. oscas1-RDV) were conducted to test the RDV acquisition rate of non-viruliferous *N. virescens*. The final number of non-viruliferous vectors on WT-RDV (RDV-infected) rice plants was 30, 39, 33, 30, and 27, and on oscas1-RDV (RDV-infected) rice plants was 21, 18, 22, 20, and 24, respectively. The RDV acquisition rate of non-viruliferous *N. virescens* was 60.04% on WT-RDV and 44.05% on oscas1-RDV, and was significantly different (Figure 7A, t = 5.176, df = 8, P = 0.008). Similarly, the RDV transmission rate of viruliferous *N. virescens* fed on treated plants (WT + 2-heptanol) was 63.04% and on control rice plants (WT) was 86.86%, which was significantly different (Figure 7B, t = 9.436, df = 8, *p* < 0.001).

## 3. Discussion

The interactions among host plants, plant viruses, and insect vectors are multiple and complex [3,27,28]. Plant virus infection could alter chemical cues and behavioral changes in its host plants and insect vectors that enhance virus epidemic and dispersal [4,10,29,30]. The results from this present study support our hypothesis that EBC and 2-heptanol facilitate RDV acquisition and transmission by influencing the olfactory behavior of both non-viruliferous and viruliferous *N. virescens* independently. Firstly, non-viruliferous *N. virescens* preferred RDV-infected rice plants (odors) over RDV-free rice plants (odors), and conversely, viruliferous *N. virescens* preferred RDV-free rice plants (odors) to RDV-infected rice plants (odors). Secondly, EBC attracted non-viruliferous *N. virescens* and 2-heptanol repelled viruliferous *N. virescens*; revealed by a series of complementation assays. Thirdly, the two VOCs promote the rate of RDV acquisition and transmission by insect vectors.

Changes in host selection preference (conditional preference) by insect vectors may promote virion spreading and diffusion, especially for vectors transmitted plant viruses in persistent and circulative manners, such as BYDV and potato leaf roll virus (PLRV) transmitted by *M. persicae*, tomato spotted wilt virus (TSWV) transmitted by *Frankliniella occidentalis*, maize Iranian mosaic virus (MIMV) transmitted by *Laodelphax striatellus*, and tomato yellow leaf curl virus (TYLCV) and tomato severe rugose virus (ToSRV) transmitted by *Bemisia tabaci* [5,6,31,32,33]. In the present study, we found that RDV modified the host selection preference of the vector *N. virescens*, which was consistent with our previous studies [8,26]. These findings suggested that the two vectors of RDV showed conditional preference. According to the general epidemiological model established by Gandon [32], and Shaw et al. [34], we infer that the conditional preference of *N. virescens* would aid the RDV dispersion.

Virus-induced plant volatiles may be attractive or repulsive to insect vectors, and thus play a key role in host plant–plant virus–insect vector interactions [11]. Most research has focused on the behavioral responses of insect vectors to blended VOCs from virus-infected plants, or artificially blended VOCs, which mimic the natural blends, or individual VOCs prominent in those blends [6,12,17,35,36]. However, the mechanism by which virus-induced plant volatiles, especially individual VOCs, affect the conditional preference of both non-viruliferous and viruliferous vectors has not been much explored. Here, we show that two rice VOCs induced by RDV infection mediated the olfactory behavior of vector *N. virescens* insects in different manners: EBC was attractive to non-viruliferous *N. virescens* causing them to prefer RDV-infected plants, while 2-heptanol was repulsive to viruliferous *N. virescens* causing them to feed on RDV-free plants. This finding is consistent with the role of the two VOCs in non-viruliferous and viruliferous *N. cincticeps* [26]. However, whether the two VOCs could exacerbate the diffusion of RDV is unclear. In this study, for the first time, we revealed that EBC could facilitate the acquisition of RDV by non-viruliferous *N. virescens*, and 2-heptanol would aid the spread of RDV by viruliferous *N. virescens*.

EBC was found to be attractive to several insect vectors, such as *Nilaparvata lugens*, *Sogatella furcifera*, *N. cincticeps,* and apple psyllids [24,37,38,39], as well as the pest parasitic wasps *Cotesia marginiventris*, *Anagrus nilaparvatae*, *Peristenus spretus*, *Aphidius gifuensis,* and *Trichogramma chilonis* [36,40,41,42]. Moreover, the repellency of EBC to *Diaphorina citri*, the main Huanglongbing vector, has also been reported [43]. 2-heptanol was repellent to *N. lugens*, *Tribolium castaneum,* and *Rhyzopertha dominica* [44,45]. These findings taken together with our results suggest that EBC and 2-heptanol could be developed and used as a broad-spectrum attractant and repellent for controlling multiple agricultural pests.

In this study, we have also shown that the impacts of EBC and 2-heptanol on the olfactory behavior of *N. virescens* are conversely dependent upon whether the insect is carrying the virus or not. Several recent studies found that the odorant binding protein (OBP) genes or olfactory receptor co-receptor (Orco) gene of insect vectors are the target genes of plant pathogens to mediate the different olfactory behavior of non-infected and infected vectors [46,47,48,49,50]. Therefore, we hypothesize that the OBP genes or Orco gene of *N. virescens* may be the target genes of RDV that influence the olfactory perception of the two VOCs and the behavioral discrimination by *N. virescens*, and these complex interactions need further investigation.

In summary, we show that EBC attracts non-viruliferous *N. virescens* resulting in their preference for RDV-infected rice plants over RDV-free rice plants, and 2-heptanol repels viruliferous *N. virescens,* resulting in their preference for RDV-free rice plants rather RDV-infected rice plants. Importantly, we show, for the first time, that virus-induced VOCs contribute to virus acquisition and transmission by insect vectors. However, these investigations are all conducted in laboratory conditions, and whether the two volatiles would be used in controlling this devastating disease of rice plants need serious and rigorous field experiments to test.

## 4. Materials and Methods

### 4.1. Insects and Rice Plants

The colony of *N. virescens* was originally collected from rice fields in Xundian, China, and was maintained on susceptible rice seedlings (Taichung Native1, TN1) in nylon cages (80-mesh, 45 cm^3^) for several generations. A population of non-viruliferous or viruliferous *N. virescens* was established and maintained as previously described [8]. The population of non-viruliferous and viruliferous *N. virescens* was reared on TN1 (RDV-free) rice plants and RDV-TN1 (RDV-infected) rice plants respectively in a climate chamber at 26 ± 1 °C, 70 ± 5% relative humidity, under a regime of 14 h: 10 h (light: dark).

The wild-type (WT) (Minghui63) and oscas1 mutant (suppression of EBC synthase *OsCAS* via CRISPR-Cas9 system) [26] rice lines were used. RDV-infected rice plants were obtained and grown hydroponically in the greenhouse under natural lighting at a temperature of 26 ± 1 °C as previously described [8] for 40 d old and then used for experiments after confirming their RDV infection by the real-time (RT) PCR. The methods of obtaining viruliferous *N. virescens* and RDV-infected plants can be found in the Appendix A.

### 4.2. RDV Detection by RT-PCR

Total RNA was individually extracted from rice leaves or insects using Trizol reagent (Invitrogen, Carlsbad, CA, USA) following the manufacturer’s procedure. cDNA was synthesized with 1 μg RNA using TransScript One-Step gDNA Removal and cDNA Synthesis SuperMix (Transgen, Beijing, China). The primers were designed based on the S8 fragment of RDV [51]. Forward primer, 5′-ATAGCTGGCGTTACGGCTAC-3′; reverse primer, 5′-AAACCGTCCACCTGACTACG-3′. RT-PCR was performed in a 20 μL reaction containing 10 μL 2 × TransTaq HiFi PCR Super Mix (Transgen, Beijing, China), 2 μL cDNA template, 1 μL forward primer, 1 μL reverse primer, and 6 μL sterile H_2_O. The PCR was conducted with the following procedure: 94 °C for 3 min, 35 cycles of 94 °C for 30 s, 55 °C for 30 s, 72 °C for 2 min, and 72 °C for 10 min. The RDV infection status of rice plants or insects was confirmed by the RT-PCR results.

### 4.3. Studies on N. Virescens Feeding and Plant Odor Preferences

To understand the role of two VOCs ((E)-β-caryophyllene, EBC and 2-heptanol) in non-viruliferous and viruliferous *N. virescens*, for the first, the feeding and plant odor preference of *N. virescens* between WT (RDV-free) and WT-RDV (RDV-infected) rice plants needs to be definite. One WT and one WT-RDV rice plant were respectively planted into the device (containing Kimura B nutrient solution [52]) as previously described [8]. The device was a transparent polyethylene cylindrical cage (D = 18 cm, H = 50 cm) containing a plastic pot. Each cage contained two nylon mesh plugs (D = 5 cm) through which insects were transferred and to promote plant respiration (Appendix A). Fifteen female adults (<24 h old) were starved for 4 h and then placed into each device, respectively. The number of non-viruliferous or viruliferous insects on each plant was recorded at 2, 4, 8, 24, 48, and 72 h post-inoculation (PI). Fifteen biological replicates were run for each test group.

A four-quadrant olfactometer with four arenas (Camsonar SIM-4, Camsonar Group Limited, UK) was used to determine the effects of host plant odors on the olfactory behavior of both non-viruliferous and viruliferous *N. virescens,* as previously described [26]. Odor materials (WT, RDV-free rice plants; WT-RDV, RDV-infected rice plants; clean air) were individually transferred into a glass bottle (D = 10 cm, H = 60 cm). All devices were connected using odorless PVC tubes (TYGON, LMT−55, Tokyo, Japan). One non-viruliferous or viruliferous female adult (starved for 4 h) was placed into the olfactometer and after 5 min, the location and activity of the insect were recorded for 20 min (Appendix A). Data were analyzed using image-processing software (EthoVision XT 14, Noldus Information Technology, Wageningen, The Netherlands). A heat map was used to visualize the tracks made by the *N. virescens* in the four-odor area over the 20 min period (the same as below). If the residence time of the adult in the olfactometer was <95% of the total test time, the data were excluded. Forty-five biological replicates were carried out for each experiment.

### 4.4. The Effects of Individual VOCs on the Olfactory Behavior of N. Virescens

The effects of individual VOCs on the olfactory behavior of vectors were determined using a four-quadrant olfactometer (as described above). EBC (CAS: 87−44−5) and 2-heptanol (2-hep) (CAS: 543−49−7) were purchased from Sigma-Aldrich. The concentrations of the two VOCs used for testing (0.01 μg μL^−1^, 0.1 μg μL^−1^, 1 μg μL^−1^) were as previously described [26]. The two VOCs were dissolved in paraffin oil. Then, 10 mL solution (VOCs in paraffin oil) was placed into an odor source bottle, and 10 mL paraffin oil was placed into the other three glass bottles individually. Forty-five biological replicates for each concentration of each VOC were carried out.

### 4.5. Validating the Role of VOCs in N. Virescens with Feeding and Plant Odor Preference Studies

RDV infection could not induce more EBC to be released from oscas1-RDV rice plants [26]. Thus, the oscas1 rice plants were used to further validate the attraction of EBC to non-viruliferous *N. virescens*. The feeding and plant odor preference studies were carried out as described above. The tested combinations of rice plants were (1) WT vs. oscas1; (2) oscas1 vs. oscas1-RDV; (3) WT-RDV vs. oscas1-RDV.

To verify the repellent effects of 2-heptanol on viruliferous *N. virescens* with feeding and plant odor preference studies, the tested combination was WT rice plant (added synthetic 2- heptanol (8.17 μg) in 10 μL of lanolin paste) vs. WT rice plant (added 10 μL of pure lanolin paste on rice leaves) [26].

### 4.6. The RDV Acquisition Rate by Non-Viruliferous N. Virescens and RDV Transmission Rate by Viruliferous N. Virescens

The testing device for RDV acquisition experiments was the same as for the feeding preference study above. The RDV acquisition rate by non-viruliferous *N. virescens* was compared using two combinations of rice plants: (1) WT-RDV vs. WT and (2) oscas1 vs. oscas1-RDV. Five biological replicates using three plants of each type and forty-five insects were carried out. Whether these insects carried RDV or not was confirmed as per the previous method [8]. The RDV acquisition rate was calculated from the RT-PCR results as above.

The RDV transmission rate by viruliferous *N. virescens* was determined using a combination of rice seedlings of WT + 2-heptanol vs. WT rice plant. For each group, 48 rice plants were used (Appendix A). There were five biological replicates and each replicate contained forty-five insects. After 72 h, these insects were removed, and after a further 25 days, the RDV infection status of the rice plants was confirmed as above and the RDV transmission rate was calculated.

### 4.7. Statistical Analysis

All data were performed using SPSS 20.0 software. Feeding preference data were analyzed by Wilcoxon’s signed-ranks tests [53]. The residence time of N. virescens in the four quadrants of the olfactometer, containing plant odors and individual VOCs, were analyzed using a non-parametric test (Friedman–ANOVA, *p* < 0.05), and the differences attributed to four fields by Wilcoxon–Wilcox as a post hoc test [54]. The RDV acquisition rate by non-viruliferous *N. virescens* and the RDV transmission rate by viruliferous *N. virescens* were analyzed using the Student’s *t*-test.

## Figures and Tables

**Figure 1 ijms-24-01777-f001:**
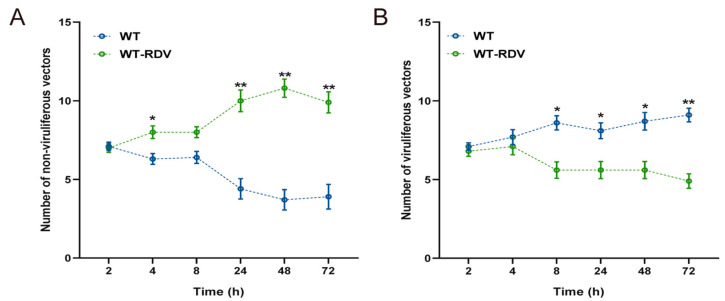
Feeding preference of non-viruliferous (**A**) and viruliferous *N. virescens* (**B**) between WT and WT-RDV rice plants. RDV, rice dwarf virus; WT, wild type (Minghui63, MH63) (RDV-free); WT-RDV, MH63-RDV (RDV-infected). All feeding choice assays were performed with fifteen biological replicates, each of the fifteen new emergent adults. The statistical differences between the two treatments in the same column were indicated by asterisks (* *p* < 0.05; ** *p* < 0.01, Wilcoxon’s signed-ranks tests). Data are mean ± standard error.

**Figure 2 ijms-24-01777-f002:**
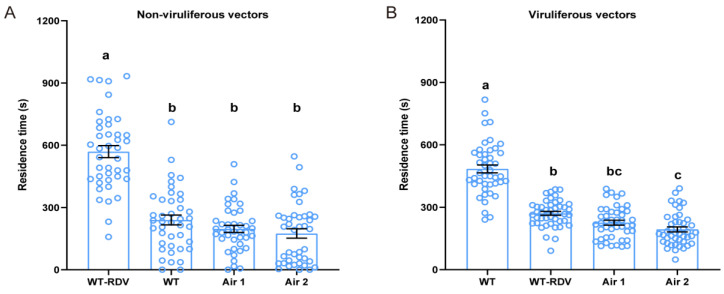
Plant odor preference of *N. virescens* exposed to different rice plant volatiles. Residence time of non-viruliferous *N. virescens* (**A**) (*n =* 41) and viruliferous *N. virescens* (**B**) (*n =* 44) in arenas permeated with WT-RDV rice plant odors and WT rice plant odors in the four-field olfactometer during the entire test time (1200 s). RDV, rice dwarf virus; WT, wild type (Minghui63, MH63) (RDV-free); WT-RDV, (MH63-RDV) (RDV-infected). WT-RDV, RDV-infected plant odors; WT, RDV-free plant odors; Air1, clean air; Air2, clean air. Deviations from equal distribution were analyzed with a Friedman–ANOVA (*p* < 0.05). Bars annotated with different letters are significantly different from each other (Wilcoxon–Wilcox test). Data are mean ± standard error.

**Figure 3 ijms-24-01777-f003:**
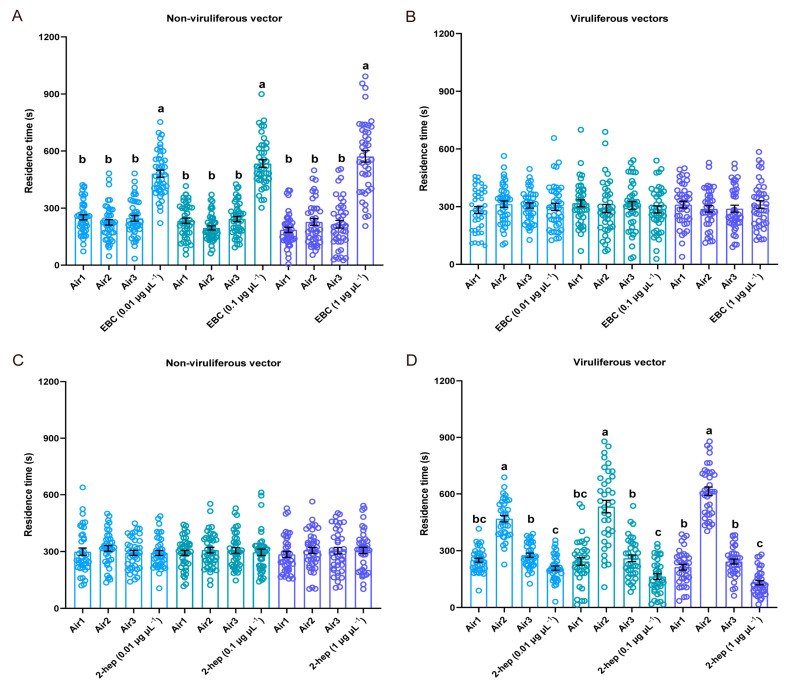
Behavioral responses of *N. virescens* when exposed to individual VOCs. Residence time in the four-field olfactometer as in Figure 2. The effects of three doses of (E)-β-caryophyllene on non-viruliferous *N. virescens* (**A**) (0.01 μg μL^−1^, *n =* 42; 0.1 μg μL^−1^, *n =* 42; 1 μg μL^−1^, *n =* 42) and viruliferous *N. virescens* (**B**) (0.01 μg μL^−1^, *n =* 40; 0.1 μg μL^−1^, *n =* 40; 1 μg μL^−1^, *n =* 39). The effects of three doses of 2-heptanol on non-viruliferous *N. virescens* (**C**) (0.01 μg μL^−1^, *n =* 38; 0.1 μg μL^−1^, *n =* 40; 1 μg μL^−1^, *n =* 41) and viruliferous *N. virescens* (**D**) (0.01 μg μL^−1^, *n =* 36; 0.1 μg μL^−1^, *n =* 36; 1 μg μL^−1^, *n =* 37). EBC, (E)-β-caryophyllene odors; 2-heptanol, 2-heptanol odors; Air1, clean air; Air2, clean air; Air3, clean air. Deviations from equal distributions were analyzed with a Friedman–ANOVA (*p* < 0.05). Bars annotated with different letters are significantly different from each other (Wilcoxon–Wilcox test). Data are mean ± standard error.

**Figure 4 ijms-24-01777-f004:**
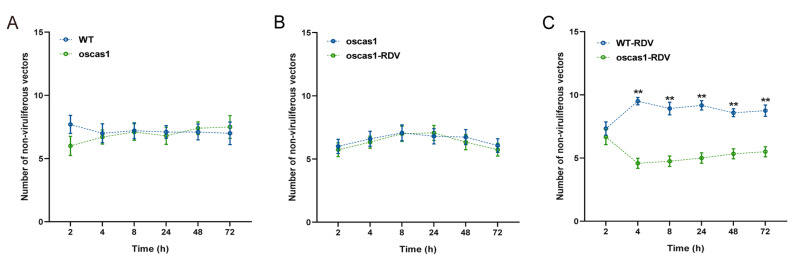
Feeding preferences of non-viruliferous *N. virescens* between WT and oscas1 rice plants (**A**); oscas1 and oscas1-RDV rice plants (**B**); and WT-RDV and oscas1-RDV rice plants (**C**). RDV, rice dwarf virus; WT, wild type (Minghui63, MH63) (RDV-free); WT-RDV, MH63-RDV (RDV-infected); oscas1, mutant rice plant (RDV-free); oscas1-RDV, mutant rice plant (RDV-infected). All feeding choice assays were performed with fifteen biological replicates, each of fifteen newly emerged adults. The statistical differences between the two treatments in the same column were indicated by asterisks (** *p* < 0.01, Wilcoxon’s signed-ranks tests). Data are mean ± standard error.

**Figure 5 ijms-24-01777-f005:**
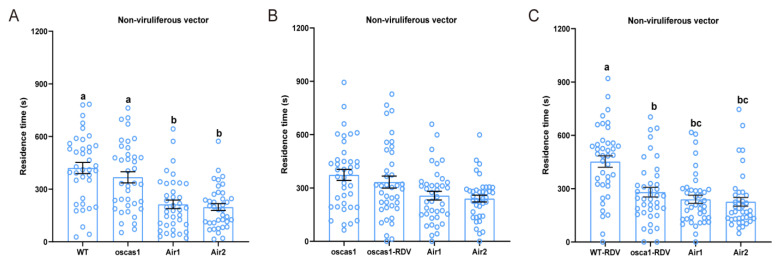
Plant odor preferences of non-viruliferous *N. virescens* exposed to WT and oscas1 rice plants (**A**) (*n =* 41); oscas1 and oscas1-RDV rice plants (**B**) (*n =* 39); and WT-RDV and oscas1-RDV rice plants (**C**) (*n =* 40). RDV, rice dwarf virus; WT, wild type (Minghui63, MH63) (RDV-free); WT-RDV, MH63-RDV (RDV-infected); oscas1, mutant rice plant (RDV-free); oscas1-RDV, mutant rice plant (RDV-infected). WT, WT plant odors; oscas1, oscas1 plant odors; oscas1-RDV, oscas1-RDV plant odors; WT-RDV, WT-RDV plant odors; Air1, clean air; Air2, clean air. Deviations from equal distribution were analyzed with a Friedman–ANOVA (*p* < 0.05). Bars annotated with different letters are significantly different from each other (Wilcoxon–Wilcox test). Data are mean ± standard error.

**Figure 6 ijms-24-01777-f006:**
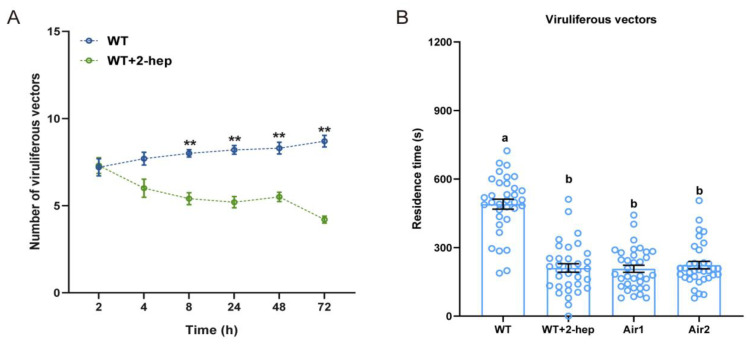
The effects of 2-heptanol sprayed on rice plants on the feeding and plant odor preferences of viruliferous *N. virescens*: (**A**) Feeding preference of viruliferous *N. virescens* between WT plants plus 10 μL pure lanolin paste (control plants) and WT plants plus synthetic 2-heptanol (2-hep, 8.17 μg) in 10 μL of lanolin (treated plants). WT, wild type (Minghui63, MH63) (RDV-free). All feeding choice assays were performed with fifteen biological replicates, each of fifteen newly emerged adults. The statistical difference between the two treatments in the same column is indicated by asterisks (** *p* < 0.01, Wilcoxon’s signed-ranks tests). Data are mean ± standard error. (**B**) Plant odor preference of viruliferous *N. virescens* between control plants and treated plants (*n =* 34). WT, WT plant odors; WT + 2-hep, WT plants plus synthetic 2-heptanol odors; Air1, clean air; Air2, clean air. Deviations from equal distributions were analyzed with a Friedman–ANOVA (*p* < 0.05). Bars annotated with different letters are significantly different from each other (Wilcoxon–Wilcox test). Data are mean ± standard error.

**Figure 7 ijms-24-01777-f007:**
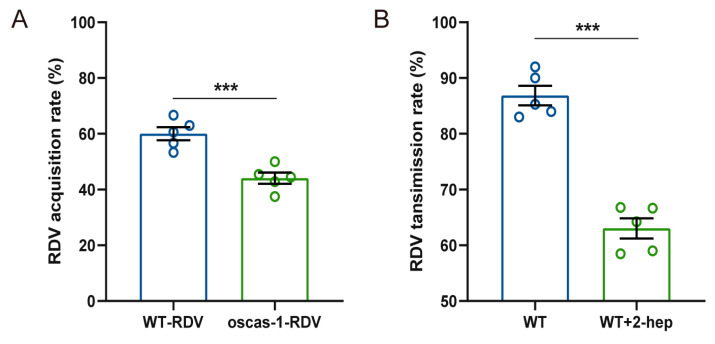
RDV acquisition and transmission rate by *N. virescens*: (**A**) RDV acquisition rate by non-viruliferous *N. virescens* between WT and WT-RDV rice plants, and between oscas1 and oscas1-RDV rice plants. (**B**) RDV transmission rate by viruliferous *N. virescens* between WT plants plus 10 μL pure lanolin paste (WT) and WT plants plus synthetic 2-heptanol (2-hep, 8.17 μg) in 10 μL of lanolin (WT+2-hep). WT, wild type (Minghui63, MH63) (RDV-free); WT-RDV (MH63-RDV) (RDV-infected); oscas1, mutant rice plant (RDV-free); oscas1-RDV, mutant rice plant (RDV-infected). (*** *p* < 0.01, Student’s *t*-test). Data are mean ± standard error (*n =* 5).

## Data Availability

Data are contained within the article or Appendix A.

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
