# Peer review of "Virus-Induced Plant Volatiles Promote Virus Acquisition and Transmission by Insect Vectors"

_ijms, 2023, doi:10.3390/ijms24021777_

Round 1

Reviewer 1 Report

Dear authors,

I have reviewed your manuscript " Virus-Induced Plant Volatiles Promote Virus Acquisition and Transmission by Insect Vectors" submitted for publication in International Journal of Molecular Sciences. The manuscript is interesting and presents a valuable collection of information on virus-vector-plant interactions using rice dwarf virus (RDV) and insect vector Nephotettix virescens as a model. Additionally, this manuscript suggested prevention and control strategies for the disease management.

I have highlighted a few points below that could improve the quality of the manuscript before publication.

I suggest that all procedures be described in the material and methods rather than referencing other papers.

Please describe all abbreviations in figures, such as “WT”, “WT-RDV”, “RDV”, etc.

Line 27: for the disease management.

Line 33: plant viruses, and

Line 46: aromatics, and

Line 91: 0.041; 48 h, P = 0.024; 72 h, P = 0.006) (Fig. 1B).

Line 232: plant viruses, and insect

Lines 253-254: Gandon [30] and Shaw et al. [32]

Line 302: Consider briefly adding how the insect were kept.

Line 305: Consider briefly adding the conditions in which plants were kept.

Line 306: How was RDV infection confirmed? Add this information and describe the method.

Lines 313-314: Yoshida et al. 1976? Please add this reference to the reference list and fix the format of the reference in the text.

Line 314: Briefly describe the procedure.

Lines 359-360: Please describe the procedure so others can replicate this. How about primers? 

Author Response

Comments and Suggestions for Authors

Dear authors,

I have reviewed your manuscript " Virus-Induced Plant Volatiles Promote Virus Acquisition and Transmission by Insect Vectors" submitted for publication in International Journal of Molecular Sciences. The manuscript is interesting and presents a valuable collection of information on virus-vector-plant interactions using rice dwarf virus (RDV) and insect vector Nephotettix virescens as a model. Additionally, this manuscript suggested prevention and control strategies for the disease management.

Response: We thank the reviewer for finding our work well executed and communicated.

I have highlighted a few points below that could improve the quality of the manuscript before publication.

I suggest that all procedures be described in the material and methods rather than referencing other papers.

Response: We have perfected relevant procedures.

Please describe all abbreviations in figures, such as “WT”, “WT-RDV”, “RDV”, etc.

Response: We have described all abbreviations in figures.

Line 27: for the disease management.

Response: We have revised this as suggested.

Line 33: plant viruses, and

Response: This has been amended.

Line 46: aromatics, and

Response: This has been amended.

Line 91: 0.041; 48 h, P = 0.024; 72 h, P = 0.006) (Fig. 1B).

Response: This has been amended.

Line 232: plant viruses, and insect

Response: This has been amended.

Lines 253-254: Gandon [30] and Shaw et al. [32]

Response: We have revised this as suggested.

Line 302: Consider briefly adding how the insect were kept.

Response: We have added the conditions. See L342-345.

Line 305: Consider briefly adding the conditions in which plants were kept.

Response: We have added the conditions. See L348-349.

Line 306: How was RDV infection confirmed? Add this information and describe the method.

Response: We have added the method. See L353-364.

Lines 313-314: Yoshida et al. 1976? Please add this reference to the reference list and fix the format of the reference in the text.

Response: We have revised this as suggested.

Line 314: Briefly describe the procedure.

Response: We have added relevant information in L372-374.

Lines 359-360: Please describe the procedure so others can replicate this. How about primers? 

Response: We have described the procedure in line 353-364.

Reviewer 2 Report

This is a very interesting study that reports and confirms the specific role of two plant volatiles induced by viral infection to promote the spread of a virus. Their claims appear to be adequately proven and their results suppose an important contribution to the field. The experimental design, number of replicates and statistical analysis are adequate to prove their hypothesis.

My main concern regards the clarity and redaction of the manuscript. The English language and style of the manuscript must be carefully revised as there are mistakes and poor grammar usage, which even difficult the understanding of the results in some cases. My suggestions (there are more minor mistakes that should be changed but due to the limited period to review this manuscript offered by the journal I lack the time to detail everything, but I encourage a thorough revision by the authors) are the following:

The names of virus are incorrectly written according to the ICTV rules (A virus name should never be italicized, even when it includes the name of a host species or genus, and should be written in lower case. This ensures that it is distinguishable from a species name, which otherwise might be identical.) Examples:

-Line 40 Barley yellow dwarf virus (BYDV) should be barley yellow dwarf virus (BYDV)

-Lines 50-51 by Cucumber mosaic virus (CMV), BYDV and Potato leaf roll virus (PLRV) should be by cucumber mosaic virus (CMV), BYDV and potato leaf roll virus (PLRV)

-Line 57 and Tomato yellow leaf curl China virus (TYLCCNV) inhibit the should be and tomato yellow leaf curl China virus (TYLCCNV) inhibit the

Wrong English usage:

-Line 16 Herein, worked with rice could be Herein, we worked with rice

-Line 27 promote to design new prevention and control strategies for the disease control could be promote the development of new prevention and control strategies for disease management

-Lines 67-68 In previous study, results showed that (E)-β-caryophyllene (EBC) and 2-heptanol induced by virus infection should be In a previous study, the results showed that (E)-β-caryophyllene (EBC) and 2-heptanol were induced by virus infection

-Line 240-241 2-heptanol is wrongly spelled

-Line 307 could be found should be replaced by can be found

-Line 365 d should be replaced by days

Suggestions about the style:

-Lines 75-76 This study extends instead of This study will extend

-Line 98 To verify the involvement of the rice volatiles instead of To verify the rice volatiles

-Use bioreplicate instead of replications

- were applied via two opposite arms of a four chamber olfactometer (see material and methods for details) instead of were applied via two opposite arms of the olfactometer.

- An introductory sentence is missing to provide the necessary context of the results between lines 124-125 and lines 216-217

-Line 294 in lab should be replaced by in laboratory conditions

Additionally, it would have been nicer if the authors include a last figure as a graphical abstract (if having 8 figures is allowed by the journal) to better represent in a more visual way the conclusions of this research.

Author Response

Comments and Suggestions for Authors

This is a very interesting study that reports and confirms the specific role of two plant volatiles induced by viral infection to promote the spread of a virus. Their claims appear to be adequately proven and their results suppose an important contribution to the field. The experimental design, number of replicates and statistical analysis are adequate to prove their hypothesis.

Response: We thank the reviewer for assessing our work as thorough and beautifully presented.

My main concern regards the clarity and redaction of the manuscript. The English language and style of the manuscript must be carefully revised as there are mistakes and poor grammar usage, which even difficult the understanding of the results in some cases. My suggestions (there are more minor mistakes that should be changed but due to the limited period to review this manuscript offered by the journal I lack the time to detail everything, but I encourage a thorough revision by the authors) are the following:

Response: The English language and style of the manuscript has been improved by a native English-speaking staff.

The names of virus are incorrectly written according to the ICTV rules (A virus name should never be italicized, even when it includes the name of a host species or genus, and should be written in lower case. This ensures that it is distinguishable from a species name, which otherwise might be identical.) Examples:

-Line 40 Barley yellow dwarf virus (BYDV) should be barley yellow dwarf virus (BYDV)

Response: We have revised this as suggested.

-Lines 50-51 by Cucumber mosaic virus (CMV), BYDV and Potato leaf roll virus (PLRV) should be by cucumber mosaic virus (CMV), BYDV and potato leaf roll virus (PLRV)

Response: We have revised this as suggested.

-Line 57 and Tomato yellow leaf curl China virus (TYLCCNV) inhibit the should be and tomato yellow leaf curl China virus (TYLCCNV) inhibit the

Response: We have revised this as suggested.

Wrong English usage:

-Line 16 Herein, worked with rice could be Herein, we worked with rice

Response: We have revised the abstract. See L18-19.

-Line 27 promote to design new prevention and control strategies for the disease control could be promote the development of new prevention and control strategies for disease management

Response: We have revised this as suggested. See L34-35.

-Lines 67-68 In previous study, results showed that (E)-β-caryophyllene (EBC) and 2-heptanol induced by virus infection should be In a previous study, the results showed that (E)-β-caryophyllene (EBC) and 2-heptanol were induced by virus infection

Response: We have revised this as suggested: In our previous study, we showed that EBC and 2-heptanol, induced by RDV infection, independently influenced the olfactory behavior of non-viruliferous and viruliferous N. cincticeps. See L82-84.

-Line 240-241 2-heptanol is wrongly spelled

Response: This has been amended.

-Line 307 could be found should be replaced by can be found

Response: We have revised this as suggested.

-Line 365 d should be replaced by days

Response: This has been amended.

Suggestions about the style:

-Lines 75-76 This study extends instead of This study will extend

Response: We have revised this as suggested.

-Line 98 To verify the involvement of the rice volatiles instead of To verify the rice volatiles

Response: We have revised this as suggested.

-Use bioreplicate instead of replications

Response: We have revised “replications” to “biological replicates”.

- were applied via two opposite arms of a four chamber olfactometer (see material and methods for details) instead of were applied via two opposite arms of the olfactometer.

Response: We have revised this as suggested.

- An introductory sentence is missing to provide the necessary context of the results between lines 124-125 and lines 216-217

Response: We have revised it. See L149-152 and L251-252.

-Line 294 in lab should be replaced by in laboratory conditions

Response: We have revised this as suggested.

Additionally, it would have been nicer if the authors include a last figure as a graphical abstract (if having 8 figures is allowed by the journal) to better represent in a more visual way the conclusions of this research.

Response: We have provided a graphical abstract figure but not as figure 8.

Reviewer 3 Report

The originality and fixation of your work have been found to be acceptable. However, the purpose and method explanations are insufficient. Based on this main point of view, I have given some of my comments and suggestions for your paper. Please fin them in an attachment file. 

I wish you a new year full of science

Author Response

We are very thankful to you for the careful and constructive criticism and suggestions for improvement of our original manuscript.

The abstract should reflect a complete summary of the work. Only the result is mentioned. Why such a study was needed, why this host, pathogen, and vector were studied, and the material method should have been mentioned in a few sentences in the abstract.

Response: We have revised the abstract to better present our work. See L13-36.

In the Introduction, the general subject is mentioned directly, but it is not specified. In other words, the importance of the studied virus and its vector was not mentioned. It was not explained why the rice dwarf virus (RDV) the insect vector Nephotettix virescens (Hemiptera: Cicadellidae) was studied.

Response: We have added the importance of the studied virus and its vector. See L74-82.

In L84-85, “no clear feeding preference was found between the two different rice plants at 2 h and 8 h postinoculation (PI), but there was a small significant at 4 h PI (2 h, P = 0.902; 4 h, P = 0.046; 8 h, P = 0.064). However, it was tried to be mentioned that there was a slight difference in 4h as far as I understood, this seemed a little illogical to me. Was the sentence worded incorrectly.

Response: We have revised the sentence for better understanding. See L100-103.

In L84, the situation in 2 different rice plants is mentioned, but figure 1 is explained as if there is only one hungry rice plant.

Response: The two different rice plants were WT (RDV-free) and WT-RDV (RDV-infected) rice plants. The hungry is for the tested insects (not rice plants). The two different rice plants were in one feeding device (Figure S5).

In lL133, “virescens, but had no effect towards on viruliferous N. virescens.” “on” should be better instead of “towards”

Response: We have revised this as suggested.

In L143 “Figure 3. Behavioral of N. virescens” “l” letter should be added.

Response: We have revised this as suggested.

In L167-168, “The statistical differences between two treatments in the same column are indicated by asterisks (* P < 0.05; ** P < 0.01, Wilcoxon’s signed-ranks tests).”

Response: We have revised this as suggested.

In L173  “in the olfactometer was significantly different….. “

Response: This has been amended.

In L176-177 “Non-viruliferous vectors walked more in the WT-RDV plants odor arena compared to the other three odor arenas …..”

Response: This has been amended.

Figure captions are much more than normal text. What is intended to be explained in the figure captions should be explained in the relevant paragraph. This is particularly the case with figures 3 and 6.

Response: Many thanks, figure 3 and 6 are the combinated assays. Thus their captions are more longer than other figure caption. We do our best to compress the annotations. And we have added relevant information in the relevant paragraph.

In L241 “….revealed by a series of complementation assays.” "serious" should probably be series.

Response: This has been amended.

In L250 “In the present study, we found that RDV modified the host ..”

Response: This has been amended.

In the conclusion (last) paragraph of the Discussion (L289-296), both present perfect and simple present tense was used in the same phenomenon of time. My mother language is not English, but these sentences did not suit me very well in terms of tense.

Response: Now, we all use present tense in this paragraph.

In L290, 2-hepranol should be heptanol.

Response: This has been amended.

In L294, “… all conducted in the lab, …” “the” is better to be added.

Response: This has been amended.

In L295, “…rice plants need serious and rigorous field experi..” should be corrected as it is.

Response: This has been amended.

In L 302, addition of a few sentences will be better instead of “…. as previously described [8].” 

Response: The method of obtaining viruliferous N. virescens could be found in the supplemental information.

In L 307, “… viruliferous N. virescens and RDV-infected plants could be found in the supple-..” should be corrected as it is.

Response: This has been amended.

In L309 “Studies on N. virescens feeding and plant odor preferences” is recommended for 4.2. subtitle.

Response: This has been amended.

In L314, the expressed (Fig. S1) in the Supplementary file is not very clear. Meanwhile, it would be much better if real photos were used rather than schematics.

Response: We have perfected the figure S5 to better understand.

In L332-334 sentence had better be changed to “…, a four-quadrant olfactometer should first be used to determine the effects of standard VOCs on the olfactory behavior of N. virescens (as described above).”

Response: We have revised and transferred it to L149-152.

In L 343-351, (4.4.) The studies of "validating the role of VOCs in N. virescens with feeding and plant odor preference" were not very revealing.

Response: Many thanks, in this study, we validate the role of VOCs in N. virescens with feeding and plant odor preference with a series of combined assays. And we will conduct field experiments to further verify the role of VOCs in N. virescens in future.

In L 359, “ .. the previous method [8]. “ “the” should be added before “previous”

Response: This has been amended.

In L 362, “..a combination of rice seedlings…” “a” should be added before “combination”

Response: This has been amended.

In L 363, in addition to (Fig. S2), a real photo would be better.

Response: We have given one real photo in figure S7 for better understanding.

Round 2

Reviewer 3 Report

It seems that you have taken almost all of my comments and suggestions into consideration. Thank you.